# The Association between Bullying Victimization and Subjective Well-Being among Children: Does the Role of Child Religiosity Matter?

**DOI:** 10.3390/ijerph19159644

**Published:** 2022-08-05

**Authors:** Adeem Ahmad Massarwi, Daphna Gross-Manos

**Affiliations:** 1Department of Social Work, Ben-Gurion University of the Negev, Be’er Sheva 653, Israel; 2Department of Social Work, Tel-Hai College, Upper Galilee 1220800, Israel

**Keywords:** religiosity, children, well-being, bullying victimization

## Abstract

Bullying is a major risk factor for poor psychological development for both children and adolescents worldwide. The current study, based on data from the International Survey of Children’s Well-Being (ISCWeB), explores the association between bullying victimization and subjective well-being as well as the moderating role of the child religiosity in this context among a sample of 2733 children aged 10–12 years old in Israel. Data was collected from children using self-reporting questionnaires, adopting a random stratified sampling method. A PROCESS moderation analysis was performed using SPSS for assessing the part played by child religiosity in moderating bullying victimization and the subjective well-being of children. This produced two key findings: first, children’s subjective well-being is negatively associated with bullying victimization; second, children’s religiosity serves as a protective factor by moderating the association between the child’s bullying victimization and subjective well-being. In light of the results, we recommend professionals who work with children to incorporate spiritual and religious resources into school-based interventions aiming at strengthening the child’s inner resilience and help overcome difficulties in their lives, based on a religious coping strategies.

## 1. Introduction 

### 1.1. Subjective Well-Being among Children 

Children’s subjective well-being is a substantive factor in their psychosocial development and includes cognitive and emotional aspects that reflect the children’s perceptions of their life circumstances and satisfaction [1]. In the past decade, there has been extensive interest in exploring and understanding the factors that shape children’s subjective well-being [2,3,4]. There is growing evidence that children’s well-being is affected by range of factors, including family relationships, peer relationships, and school environment [5].

Despite the importance of children’s well-being, previous studies relied mainly on adult perceptions of it in addition to focusing on objective descriptions instead of exploring children’s own subjective experience [6,7]. Furthermore, most studies have focused on well-being among adults and adolescents, with less known about subjective well-being among younger children. 

The current study is based on the International Survey of Children’s Well-Being (ISCWeB), which explored the subjective well-being of over 54,000 children from 16 countries in the world. This study explores the association between victimization by peer bullying and subjective well-being among 10–12-year-olds in Israel and the moderating role of the child religiosity on this. The current study focuses on the Israeli sample only and includes 2733 Arab and Jewish young children.

### 1.2. Bullying Victimization and Subjective Well-Being among Children

Bullying victimization can prove a major risk factor in poor psychological development for both children and adolescents worldwide [8,9,10,11]. Bullying has been defined as intentional and repeated aggression engaged in by individuals or peer groups holding greater power than does the victim [12]. Bullying can take the form of physical aggression (pushing and shoving, beating and intimidation) or verbal abuse. It can also present in relational form with bullies spreading rumors or speaking negatively about the victim behind their backs. Child victims may experience bullying in many contexts, including from peers at school [13,14]. In this study, we focus on three indicators of bullying victimization by peer at school that includes physical bullying, verbal bullying, and social bullying by exclusion. 

There is an extensive body of knowledge about how bullying victimization can affect a child’s mental health and psychosocial development. Research has shown that bullying victimization is linked to a wide range of mental health problems, such as depression and anxiety [11,15], and can have adverse behavioral outcomes including hyperactivity and involvement in antisocial conduct [16,17]. For example, a study conducted among 505 Israeli adolescents showed that bullying victimization was associated with depressive symptoms and suicide ideation [18]. 

Since children spend so much of their time at school, bullying victimization at school contributes considerably to children’s levels of subjective well-being [19,20]. Research has found that children enjoying good relations with their peers are more likely to evidence higher levels of subjective well-being (SWB) and life satisfaction [21]. Conversely, children who experience hostile relationships and aggressive interactions with their peers are more likely to indicate low levels of subjective well-being [22]. For example, a study conducted among 12,058 15-year-olds in China found that bullying victimization at school was associated with lower levels of SWB and life satisfaction [20]. Further evidence from longitudinal studies has clearly shown that being a victim of bullying has negative consequences on different aspects of schoolchildren’s lives including reduced well-being [23,24]. For example, a longitudinal study conducted among a cohort of 2232 primary and early secondary school children found that bullying victimization is a major risk factor in predicting poor mental health and socio-emotional difficulties [25]. 

Despite this extensive body of knowledge on the impact of bullying victimization on children’s well-being, most studies have focused on adolescents [20,26], with few concentrating on SWB among young children. One study conducted among 8–12-year-old schoolchildren from sub-Saharan Africa, southern Asia, and western Europe indicated that bullying victimization has a negative effect on child life satisfaction. Children who experienced physical bullying and social exclusion from other children reported lower levels of SWB than those not exposed to this [1]. 

### 1.3. Child Religiosity as a Protective Factor

In the last decade, scholars have increasingly been investigating the effects of the religious belief and spirituality on psychological development among children and adolescents [4,27,28,29]. Religiosity has been defined as encompassing a number of dimensions associated with spirituality and religious beliefs, such as believing in higher power, religious practices and involvements, such as prayer, and the importance of faith in the individual’s life [30]. The current study examines religiosity among children by exploring their beliefs in God and higher power, the frequency of their attendance at religious services and places, and the extent to which a children’s religiosity might serve as a protective factor by providing a buffer between bullying victimization and their subjective well-being.

Previous studies have found religiosity to be associated with various positive psychosocial outcomes among children and adolescents, including well-being and happiness [31,32,33,34]. It has been found that religiosity among children serves as an internal resource of finding comfort, making sense of stressful situations, and creating meaning, which assists them in coping with difficulties in their lives [35]. For example, a qualitative study conducted among 20 orphaned Ghanaian children found that self-reported child religiosity promotes well-being by encouraging positive emotions, fostering resilience, and an optimistic view of life [36]. Another study conducted among nine Iranian young children (aged 8–12), who suffered from chronic illness, found that self-reported religiosity and spirituality served as resources of hopefulness, which were comforting and helped children cope in a positive way with their condition by developing an internal power through their positive relationship with God [37]. Furthermore, a study of 11-year-old Christians in western Scotland found that children’s weekly church attendance was positively associated with lower levels of aggressive behavior and depression [38]. 

Notwithstanding increased knowledge about the direct association between child religiosity and psychosocial outcomes, less attention has been given to the buffer effect of religiosity on subjective well-being among young children, especially those who are victims of bullying. One study conducted among 103 adolescent Turkish immigrants in Germany found that adolescents’ religiosity serves as a protective factor among those who experience bullying by their peers at school by reducing depressive symptoms [39].

Guided by a resilience framework there are different theoretical models of resilience that explain the process of overcoming the negative effects of risk factors among children, such as bullying victimization. One of these theoretical models is the protective model where assets and emotional resources are suggested to reduce the effects of a risk factor on an outcome [40]. Following this theoretical approach, we are interested in exploring the moderating role of self-reported child religiosity on the association between bullying victimization and SWB among young children. 

Following this theoretical approach, we are supposing that religiosity among children is an internal resource that may reduce the negative effects of bullying victimization by helping them overcome it effectively. Granqvist seeks to understand the impact of religiosity on mental health from an attachment viewpoint, indicating that belief in God might function as a symbolic attachment figure that provides the assurance and emotional security to enable adjustment in face of danger or confusion [41]. Although this analogy has been made to explain the impact of religiosity on mental health among adults, we assume that a similar impact might be found among children and that religiosity might serve as a secure basis for well-being in the face of negative experiences such as bullying victimization.

This kind of more nuanced analysis was not formerly employed among children. Moreover, the Israeli context provides a multi-religious environment, and accordingly, the study sample includes Jewish, Muslim, and Christian children. Thus an important contribution of the study is addressing the lack of knowledge about religiosity among young children in non-Western cultures, as most of the previous studies were conducted among Christians in Western cultures [42].

### 1.4. Children in the Israeli Context

Children comprise 33.6% of the total population making Israel one of the youngest countries in the Organization for Economic Co-operation and Development with one of the highest fertility rates [43,44]. At the same time, and not without a connection, children in Israel are also some of the poorest amongst OECD countries with 31.7% of children living in poor households [45,46]. Israeli society has a large ethnic and religious diversity. Forty-five percent of Israel’s adult Jewish population are reported as secular, 25% conservative, 16% religious or very religious, and 14% ultra-orthodox. As for the Arab population, 57% consider themselves conservative, 31% are religious and very religious, and 11% are secular Seventy percent of children in Israel are Jewish, 22.1% Muslim, 1.4% Christian, and 1.6% Druze [43]. 

## 2. Aims and Hypotheses 

The study’s principal aim is to explore the moderating effect of child religiosity on the association between bullying victimization and subjective well-being by looking at a cohort of 10–12-year-olds in Israel. The study’s hypotheses are that (1) bullying victimization and SWB are negatively associated; and (2) the correlation between bullying victimization and SWB differs according to nature of the child’s religiosity. 

## 3. Methodology 

### 3.1. Study Design and Sample

The current study is based on a sample of children in Israel obtained from the third wave of the ISCWeB. The sample of 2733 10–12-year-olds was based on random sampling, clustering for ethnic strata (Jewish/Arab) to provide a representative sample of Arab and Jewish children in Israel. 

We used random cluster sampling combined with stratified sampling. An administrative list of all elementary schools in Israel from the Education Ministry, which included 2673 elementary schools, was used to randomly collect schools. This list of schools was divided into strata (clusters): religion sector (Jewish/Arab), geographical area (center/periphery), and educational supervision type, which was only relevant for the Jewish sector (secular/religious/ultra-orthodox). The combination of those layers created eight different strata. Overall, 93 schools were approached after they were systematically random sampled from the list (including some extra schools that were drawn later for the ultra-orthodox strata, which was less inclined to participate due to cultural reasons). Data was collected from 36 schools, reflecting 39% of the total sample, though it should be noted that if we do not include the ultra-orthodox strata schools, the participation rate is 63%. While the survey also collected data from second graders, we focused on the fourth and sixth-grade age groups as their data had fewer missing values.

### 3.2. Procedure 

The current study received the ethical permission of two authorities: the review board of the Hebrew University and the Ministry of Education in Israel. After ethical permissions were obtained, school principals were approached and asked to take part in the study. To encourage schools to participate we offered symbolic gifts. In schools where the principals agreed, we obtained passive agreement from the parents and active consent from the children. During data collection, children were informed by the research assistants that they are not obligated to participate if they don’t want to and can also choose which questions to answer even after starting. Self-report questionnaires were administered in classrooms by research assistants and averaged 30–50 min to complete by children base on their only perspectives, in Arabic or Hebrew. Children were free to withdraw at any time for any reason. 

### 3.3. Measurements

Child SWB was assessed using two scales that measured two main domains of well-being: cognitive well-being and life satisfaction. The scale for measuring cognitive used Children’s Worlds Subjective Well-Being Scale (SWBS) [47,48] based on the Student Life Satisfaction Scale, originally developed by Huebner [49]. The scale included six items (α = 0.936). Children were asked to indicate to what extent they agree with the statements about themselves and their lives, such as: “I enjoy my life”, “My life is going well”, “The things that happen in my life are excellent,” and so on. Responses ranged from 0 (“do not agree at all”) to 10 (“totally agree”).

In addition to the cognitive subjective well-being measurement, children were asked to indicate how satisfied they are with several aspects of their lives based using the Children’s Worlds Domain Based Subjective Well-Being Scale (DBSWB). This scale is based on the “brief multidimensional student life satisfaction scale” by Seligson, Huebner, & Valois [50]. The scale consists of five items measuring domain-based cognitive SWB to which respondents are asked to mark their level of satisfaction on an 11-point scale, from 0 (“not at all satisfied”) to 10 (“completely satisfied”). The items used were: “How satisfied are you with the people with whom you live?”, “How satisfied are you with your friends?”, “How satisfied are you with your life as a student?”, “How satisfied are you with the area where you live?”, and “How satisfied are you with the way you look?”. Cronbach’s alpha for the sample is 0.66. While the internal reliability is not high, this measure captures SWB differently as it is based on domains and thus its use is important. Moreover, confirmatory factor analysis demonstrated satisfying fit indices following the recommendation of Hooper et al. [51] and Hu and Bentler [52] as following: TLI = 0.97, CFI = 0.98, SRMR = 0.02, the exception being RMSEA = 0.11, which was higher than recommended (suggesting lower parsimony of the model, possibly because of high correlation between the items) as reported in Gross-Manos & Shimoni [53]. Both SWB scales were transformed into 0–100 scales based on the recommendations of Huebner and Cummins [54,55]. An overall score was arrived at by computing the sum of the various elements, higher scores reflecting higher levels of subjective well-being. 

Bullying victimization was measured using three questionnaire items (α = 0.70). Children were asked to indicate how often they experienced incidents of bullying by their peers at school based on the following questions: How often have other children in your school hit you?, How often have other children in your school called you by unkind names? How often have other children in your class made you feel left out? Their answers ranged from 0 (“never”) to 4 (“more than three times”). The authors of the study devised this measurement. A single comprehensive score was derived by computing the sum of the various elements, higher scores reflecting higher levels of bullying victimization by others. 

Child religiosity was measured using 4 items (α = 0.78) based on the Brief Multidimensional Measurement [42]. Items included inner dimensions of religiosity, such as: “I feel higher power presence”; “I believe in a higher power who watches over me”; “When you are worried do you depend on your religion to help you?”; “Do you think of yourself as a religious person?” Children were asked to indicate to what extent they agree with each one of these statements. Responses ranged from 0 (“don’t agree”) to 4 (“totally agree”). One overall score was derived by computing the sum of the items. Higher scores reflect higher levels of religiosity among children. 

The children were also asked to provide information about their sociodemographic characteristics: age, grade, gender, family structure, place of dwelling, and religion affiliation. 

All variables from the study were measured using reliable and valid measurements translated into Arabic and Hebrew. 

### 3.4. Data Analysis 

Descriptive statistics related to the dependent variable (child SWB), the independent and moderated variables (child religiosity and bullying by peers at school), and the control variables (child age and gender) were examined first. Following this, bivariate analyses testing the relationships between the research variables using Pearson’s correlations were performed (see Table 1). Thirdly, a moderation analysis was carried out using SPSS PROCESS-Model #1 developed by Preacher & Hayes [56] to evaluate the moderating role played by child religiosity on the relationship between child bullying victimization and child subjective well-being (see Table 2 and Table 3). In this analysis, child age and gender were held as covariates.

The SWB measures showed a skewed distribution, which is typical for SWB measures [57]. This departure from normal distribution was handled by a bootstrapping procedure (with 5000 re-sampling) that reduces the impact of anomalies and outliers. However, results with a bootstrapping procedure showed only very minor changes.

## 4. Results

### 4.1. Descriptive Statistics

The study sample included 2733 children, with slightly higher sample of fourth-grade children (N = 1429, 52.3%). The mean age was 10.62 (SD = 1.14), almost equal percentage of females (50.3%) and males (49.7%); 28.8% were Arabs, 71.2% were Jews; 5.5% of the children were not born in Israel. The study found that 17.9% of the children reported having been called unkind names by other children in their school more than three times, 10.2% reported that they had been hit by other children more than three times, and 7.6% reported they had been left out by other children in school more than three times. 

### 4.2. Bivariate Analyses

Table 1 shows that bullying victimization is negatively associated with all measurements of child subject well-being: DBSWB (*r* = −0.303, *p* < 0.001) and SWBS (*r* = −0.273, *p* <0.001). The more the child has experienced bullying by peers as a victim, the lower the level of subjective well-being reported.

As for the association between child religiosity and all measurements of SWB, the findings show significantly positive associations: DBSWB (*r* = 0.172, *p* < 0.001) and SWBS (*r* = 0.148, *p* <0.001). Any higher level of child religiosity corresponded to a higher SWB level. There were no significant correlations between child age, gender, and SWB (See Table 1).

### 4.3. The Moderating Role of Child Religiosity

Table 2 and Table 3 show the summary results for regression models for SWBS and DBSWB, presenting standardized coefficients after controlling for age and gender. The findings reported in Table 2 and illustrated in Figure 1 show that the correlation between bullying victimization and DBSW is stronger among those who reported lower levels of religiosity (*β* = 0.067, *p* =.000, 95% CI [0.02, 0.11]). This model explains 15% of the DBSWB variance. In a similar vein, the findings reported in Table 3 and illustrated in Figure 2 show that the correlation between bullying victimization and DBSW is stronger among those who reported lower levels of religiosity (*β* = 0.101, *p* = 0.000, 95% CI [0.06, 0.14]). This model explains 12% of the SWBS variance. Overall, the findings of the study show that child religiosity serves as a moderating factor in the association between bullying and SWB (SWBS and DBSWB). In both models, results were significant after controlling for child age and gender.

## 5. Discussion

### 5.1. Key Findings

The study produced two key findings. The first is that children’s SWB is negatively associated with bullying victimization. The more the children experienced bullying by other children as victims, the lower their levels of SWB and life satisfaction. The second is that child religiosity serves as a protective factor by moderating the association between bullying victimization and SWB together with life satisfaction. 

### 5.2. Bullying Victimization and SWB

Our findings provide further empirical evidence of the negative correlation between bullying victimization and overall SWB and life satisfaction among young children. These results are consistent with previous studies indicating that children having experienced bullying tend to evidence poorer well-being and lower levels of life satisfaction [58,59,60]. The more children experience bullying by their peers, such as being physically hit or socially excluded, the lower their level of SWB.

We can understand this association through the stress process model [61] where one stressor, in our case bullying victimization, creates additional stressors and adverse social relationships with significant others in the child’s environment, including their relationship with their parents, peers, and teachers. In turn, this stressful situation negatively affects the child’s perceptions of themselves, their satisfaction with life, and their relationships with others. Therefore, bullying victimization can be a major risk factor that decrease children’s well-being and prevent them from enjoying a happy and fulfilling life, particularly when bullying occurs in familiar environments such as schools [14,20]. We nonetheless recommend that future studies explore in more depth the mechanisms behind the association between experiences of bullying victimization and well-being among young children. 

### 5.3. Religiosity as a Protective Factor

The current study is among the first to explore the moderating role of child religiosity on the association between bullying victimization and SWB in young children. Its findings indicate that among those reporting lower religiosity levels, there was a stronger association between bullying victimization and all SWB measures. In other words, the findings of the study provide empirical evidence that child religiosity can shield children and reduce the harmful effects of bullying on young children on their subjective well-being and life satisfaction. The findings of the study are in line with the results of previous studies finding religiosity as a protective factor in adolescents’ mental health when faced with stressful life situations [27,62].

By interpreting the results of the study in light of a resilience theoretical approach [40], we see that religiosity serves as a protective factor in the face of risk factors and negative social experiences; bullying victimization in our case. Religiosity provides internal “protection” and resilience that help children overcome the negative impact of bullying by peers. It is an emotional coping strategy that helps children cope with stressful life events better than those with lower levels of religiosity. Therefore, the harm caused to their SWB is lower. Our findings indicate that religious children are more resilient in the face of negative experiences as they can draw on their faith to maintain a positive vision of a meaningful life [63]. This coheres with studies of adolescents indicating that those with higher levels of spirituality tend to perceive their peers in a better light [64], and, therefore, tend to forgive them when they experience conflict situations since forgiveness is an aspect of religiosity [65].

From an attachment theory point of view, belief in God and higher powers, as part of the child’s religiosity, might function as a symbolic attachment that provides assurance and emotional security to enable adjustment in the face of danger or confusion [41]. In the context of negative social experiences, such as bullying by peers, this type of emotional attachment to God provides an inner source of strength by developing an image of God as protective, caring, and responsive [66]. Relying on attachment to God seems to help bullied children regulate the emotional distress they experience in their lives, helping them to internalize a positive view of life and its meaning [42]. This can help children cope with negative social experiences positively and thus lower the harm to their well-being and life satisfaction. Similar coping mechanisms were found in studies of young children who suffer from different stressful life events such as orphanhood, chronic illness, and disability, where children’s religious beliefs helped them to cope effectively [37,67]. 

Furthermore, the findings of the current study underline that, like adolescents and adults, young children also benefit psychologically from their religious and spiritual beliefs [63,68,69].

### 5.4. Study Limitations and Possible Future Research

The study was conducted among a large and representative sample of children, but it does have a few limitations that need noting. Firstly, as a cross-sectional design was utilized, it is not possible to make causal inferences. Future studies should, therefore, be longitudinal in design to establish causality. Secondly, we recommend that future studies explore the mechanisms underlying the impact of religiosity on SWB among young children. Thirdly, we recommend that future studies collect additional information from significant informants, such as parents and teachers, as we restricted ourselves to children’s self-reporting to measure all research variables. In addition, further rigorous research is needed to better understand the impact of different components and factors of religious beliefs among young children and their parents in non-Western cultures and their contribution to children’s socio-emotional development. Lastly, we recommend testing our model with children in different settings: since the study was conducted among children in a specific sociocultural context, its results cannot be generalized to apply to other such contexts. 

## 6. Conclusions

The current study is among the first to shed light on the protective role of religiosity on children’s subjective well-being in a non-Western culture. While the majority of previous studies mainly focused on religiosity among adults and adolescents, the current study focused on a new area of research by exploring the impact of religiosity on children from their own perspectives and perceptions. The findings here show that bullying victimization presents a significant risk factor that causes harm to children’s SWB and life satisfaction, indicating the importance of developing prevention and intervention programs aiming at tackling bullying among young children. These programs must aim to reduce levels of peer bullying in schools, but also raise awareness about the negative consequences of different types of bullying on children’s perceptions of themselves, their life satisfaction and quality of life, as well as helping children cope with these problems effectively. This is extremely important in Israel, where children are exposed to high levels of various types of bullying in different contexts, including schools [18].

Since the study shows that religiosity can help provide a shield against the harmful effects of bullying victimization, it is vital to develop carefully designed interventions that account for the children’s cultural context, including how central religiosity may be for them as a source of emotional support. In light of the findings, we recommend that professionals who work with children develop spiritual-based interventions that incorporate religious dimensions (such faith and belief, optimism, and giving meaning to life) aiming at strengthening the child’s inner resilience and helping overcome difficulties in their lives based on religious coping strategies. Furthermore, we recommend that school-based professionals consult with religious community leaders to explore children’s spiritual resources and facilitate their access to these resources on a personal and a community level. 

Programs that aim to enhance well-being and life satisfaction among young children should view religiosity as a resource for effective interventions, especially among those exposed to negative social experiences and environmental risk factors. It is important for practitioners who work with children to understand and evaluate children’s religious norms and values, even if these do not reflect those they (the practitioners) hold.

## Figures and Tables

**Figure 1 ijerph-19-09644-f001:**
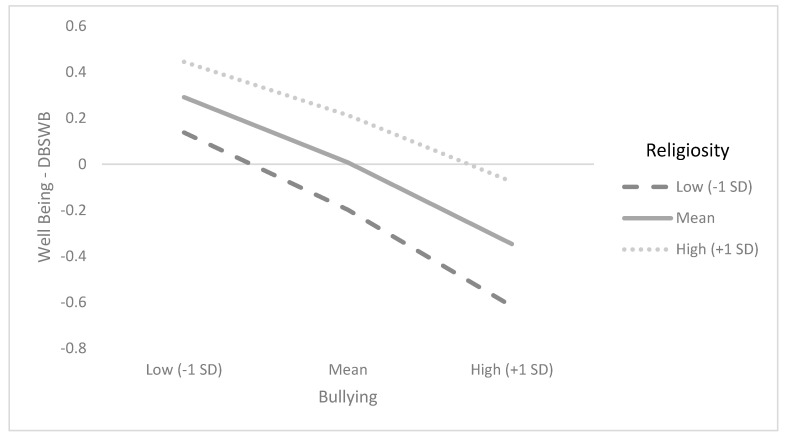
Interaction between child religiosity and bullying victimization in predicting DBSWS.

**Figure 2 ijerph-19-09644-f002:**
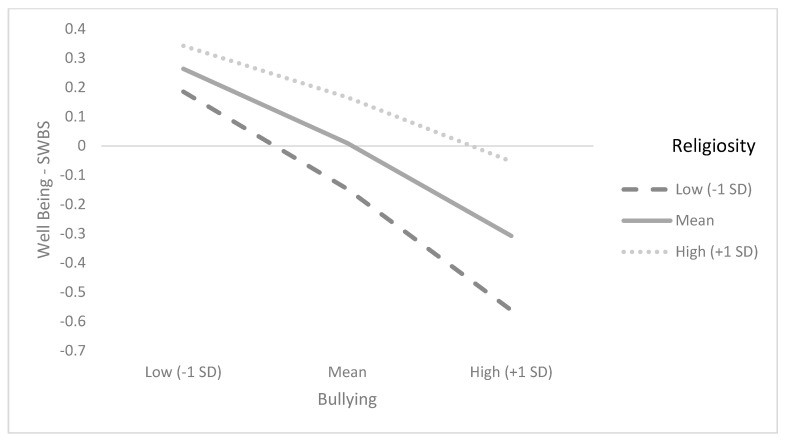
Interaction between child religiosity and bullying victimization in predicting SWBS.

**Table 1 ijerph-19-09644-t001:** Means, standard deviations and Pearson correlations of the study’s variables (N = 2733).

Variable	Mean	SD	1	2	3	4	5	6
1. Child subjective wellbeing (DBSWB)	87.38	14.23	1					
2. Child-subjective well-being (SWBS)	89.51	18.48	0.690 *	1				
3. Child religiosity	2.35	1.46	0.172 *	0.148 *	1			
4. Bullying victimization	1.94	2.35	−0.303 *	−0.273 *	−0.012	1		
5. Gender (Male = 1)	1.51	0.5	0.004	−0.002	0.036	0.080 *	1	
6. Age	10.61	1.41	−0.011	−0.019	0.011	−0.173 *	0.017	1

Note. SWBS refers to Subjective Well-Being Scale; DBSWB Refers to Domain Based Subjective Well-Being Scale; * *p* < 0.001.

**Table 2 ijerph-19-09644-t002:** Regression results of the moderation model predicting child’s subjective well-being (Domain Based Subjective Well-Being Scale—DBSWB) (N = 2733).

	Direct and Interaction Effects			
Variable	B	SE	t	*p*
Child’s religiosity	0.21	0.02	10.69	0
Bullying victimization	−0.36	0.02	−17.59	0
Age	−0.04	0.02	−2.22	0.05
Gender (male = 1)	−0.02	0.02	−0.94	n.s.
Child’s religiosity X bullying victimization	0.07	0.02	3.29	0.001

**Table 3 ijerph-19-09644-t003:** Regression results of the moderation model predicting subjective well-being (Subjective Well-Being Scale—SWBS) (N = 2733).

	Direct and Interaction Effects			
Variable	B	SE	t	*p*
Child’s religiosity	0.17	0.02	8.51	0
Bullying victimization	−0.32	0.02	−15.98	0
Age	−0.06	0.02	−3.38	0.001
Gender (male = 1)	−0.02	0.02	−0.91	n.s.
Child’s religiosity X bullying victimization	0.1	0.02	5.05	0

## Data Availability

Data availably is provided upon request.

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
