# Peer review of "The Association between Bullying Victimization and Subjective Well-Being among Children: Does the Role of Child Religiosity Matter?"

_ijerph, 2022, doi:10.3390/ijerph19159644_

Round 1
Reviewer 1 Report
The study appears to be important and timely. It discuss the cultural factors in bullying and SWB issues, which have potential good implications. However, I do have several suggestions:
1. Why the authors assumed being bullied is related to well-being of children, not vice versa? Consider this is a cross-sectional study, it is nesseceary to clarify it in intruductuion (provide theoritical evidenc).
2. Most of bullying study prefer to used word "bullying victimization", instead of bullying. Author used bullying may confused readers, as it may also indicates bullying behaviors towards other peers. Please changed it.
3. The overreaching theoritical framework is absent in 1.3 and 5.3 sections. Please add related theories to support your hypothesis.
4. The authors examine the cultue factor which is commendable. I suggests authors to argues that most of previous studies was conducted in United Stats and not fully consider culture or relogion factors.
5. The manuscript should be carefully checked and proofing to meed this journal's standards. E.g., Line 270, Superfluous periods "5.3. . Religiosity as a protective factor";
Line 214 &218, p and r should be italicized with lowercase.
Table.1:Compose type is awful.
Table.2 & 3: Keep two decimal places.
Figure 1 & 2 : Pleased used solid and dotted lines in different colors. It is not clearly enough.
minor commonts:
Why author says the sample contains "over 54,000 children 40 from 16 countries." (line 40). But in Line 137, sample only remains 2,733?
What exactely did participants from? do not just report "from various areas."
Please add all α relaibility in method sections.
In limitation section, authors could added suggestion for future studies to exam more specfic religion or cultural factors. For example, in eastern culture, Collectivism is a general cultural factor, and social harmony beliefs is more specific one, which may provide more implications for research.
Author Response
Dear reviewer,
Please see the attachment - a response letter.
Regards
Authors

Reviewer 2 Report
Comments to the author:
The manuscript presents a topic of interest that can provide relevant information about the effects of religiosity on the life satisfaction of children aged 10 to 12 years, as a protective factor against bullying.
However, some methodological aspects that determine the subsequent results and the reason for performing the different tests remain unclear.
I believe that the relevance of each item should be previously analyzed using AFE and AFC, this would give consistency to the study and reflect the appropriateness of using the different questionnaires.
There is no mention of the reliability and normality of the tests.
Some additional results would be beneficial to the reader and it would be helpful to include more detail to ensure the reader fully understands these issues. I would also like to see a little more about how the results of this study could be applied to allow this to have a greater impact on the circumstances of children in Israel.
There is talk of the context and the unexpected difficulty that young people in Israel may have, little argued, this circumstance can generate gaps in the readers.
I point out that the statistical aspects of the document are difficult to interpret and I find the middle section on the instruments used, the analysis and the results shallow if the relevance of each item is not previously explored.
Here are some additional suggestions for researchers to make statistics more accessible to the lay reader, unless they are unfamiliar with statistical analysis as research.
I would like the researchers to imagine giving a talk, without slides, about their research to explain in words what they were trying to accomplish with their statistical analysis; what the results show; and how they could be used.
I am concerned that:
• A large sample is used, but the mode of selection is not explained and the implications of this are not sufficiently recognized (possibly further research could be recommended to test the findings in young people in Israel).
• that the research confirms the findings of others without clearly promoting what is new about current research or what it adds to research in the field.
• it is stated in the abstract and conclusion that the research will have application but it is not at all clear how or what aspects of the findings should be used.
Author Response

(The authors gave the same response as above.)
